# Electrolyzed Water Treatment for the Control of the Zoonotic Pathogen *Vibrio vulnificus* in Aquaculture: A One Health Perspective

**DOI:** 10.3390/microorganisms12101992

**Published:** 2024-09-30

**Authors:** Pablo Ibányez-Payá, Adolfo Blasco, José V. Ros-Lis, Belén Fouz, Carmen Amaro

**Affiliations:** 1Institute BIOTECMED, Universitat de València, 46100 Burjassot, Valencia, Spain; pablo.ibanez-paya@uv.es (P.I.-P.); belen.fouz@uv.es (B.F.); 2Institute IDM, Universitat de València, 46100 Burjassot, Valencia, Spain; adolfo.blasco@uv.es

**Keywords:** *Vibrio vulnificus*, electrolyzed water, One Health, vibriosis, hypochlorous acid, zoonotic pathogen, sustainable control measures

## Abstract

*Vibrio vulnificus* (Vv) is a bacterial pathogen native to warm and brackish water ecosystems that can cause fatal septicemia (Vv-vibriosis) in humans and various farmed fish species. From a One Health perspective, controlling Vv-vibriosis outbreaks on farms is essential not only for animal but also for human health, as it reduces the risk of Vv transmission to humans. Electrolyzed water (EW) is a sustainable control method, exhibiting transient disinfectant properties due to the formation of hypochlorous acid (HOCl). We hypothesized that EW could effectively reduce Vv populations in aquaculture facilities, preventing outbreak emergence. To test this hypothesis, survival assays in EW were conducted under varying conditions of salinity, pH, and free available chlorine (FAC). The results indicated that an intermediate concentration of FAC had a significant bactericidal effect on Vv populations regardless of the condition and tested strain. Consequently, the strategic use of EW could serve as an eco-friendly preventive and control measure against Vv-vibriosis by significantly decreasing the bacterial load in farm water.

## 1. Introduction

*Vibrio vulnificus* is an emerging zoonotic pathogen that inhabits marine and brackish water ecosystems in temperate, tropical, and subtropical zones [1]. This species is considered a biomarker of climate change because rising water temperatures are causing its poleward expansion and proliferation in coastal waters [2]. Additionally, the increase in water temperature causes an upregulation of virulence factors involved in immune system resistance, which in turn increases the virulence of outbreaks [3].

As a human pathogen, *V. vulnificus* causes sporadic severe infections (human Vv-vibriosis) in wounds exposed to seawater or after contact with diseased fish, as well as gastroenteritis following the consumption of raw seafood [1]. Both types of Vv-vibriosis can lead to sepsis and death, especially in at-risk patients [1,4]. As an animal pathogen, it causes outbreaks of hemorrhagic septicemia (fish vibriosis) that are particularly severe in farmed European eel (*Anguilla anguilla* (Linnaeus, 1758)), its most susceptible host [5]. Notably, fish Vv-vibriosis can be transmitted via water and ingestion, although water is the primary transmission route [5]. Finally, there is clear evidence of transmission of vibriosis from fish to humans after handling diseased fish in fish farms, so that controlling fish health has a direct impact on reducing the risk of transmission of vibriosis to humans [6].

The species is divided into five phylogenetic lineages plus a pathovariety, named piscis [6]. It is believed that strains from all five lineages can cause human Vv-vibriosis, but only those from the pathovar piscis can cause fish Vv-vibriosis due to the acquisition of a transferable virulence plasmid (pVir) [6]. Recently, it has been reported that pVir, initially present in some clones of lineage 2, has already been transferred to four of the five lineages of the species, a process linked to fish farms as these artificial environments can favor genetic exchange between bacteria. Therefore, the aquaculture industry is a determining factor in the recent evolution and epidemic spread of this species [6]. Additionally, recent evidence shows that pVir has been transferred to other fish pathogens such as *V. harveyi*, a marine pathogen with a broad host spectrum [6].

From a One Health perspective, controlling Vv-vibriosis outbreaks on farms would be essential not only for animal but also for human health, as it would reduce the risk of Vv-transmission to humans. Vv-vibriosis can be treated and cured with antibiotics [7,8,9]. Although the antibiotics used in human (HAa) and fish therapy differ, the use of antibiotics in aquaculture can promote the emergence of resistance to HAa through cross-resistance, particularly among antibiotics of the same family (e.g., oxytetracycline and tetracycline) [10]. Therefore, the use of antibiotics on farms should be minimized in favor of other therapeutic methods.

This study addresses this issue and suggests that one way to control infectious outbreaks at farms is to reduce the microbial load in the water, especially pathogens, in aquaculture facilities. There are already-developed methods for reducing microbial load, including physical (filtration, UV radiation, ozonization) and chemical (H_2_O_2_, peracetic acid, and chlorinated disinfectants) approaches [11], but many generate toxic residues.

Considering all the above, this study selected and tested electrolyzed water (EW) as an alternative method to control the pathogen *V. vulnificus*. EW has proven to be a highly effective bactericide in the food and healthcare industries [12,13,14,15] and offers the advantage of not generating toxic residues, making it environmentally friendly. In addition, another advantage is that it has a low generation cost [16]. Consequently, EW is a safe, eco-friendly, and cost-effective disinfectant with broad-spectrum antimicrobial action, low toxicity, and no harmful residues, making it superior to traditional methods and, therefore, adequate for aquaculture.

## 2. Materials and Methods

### 2.1. Bacterial Strains and Cultures

Representative strains of the five phylogenetic lineages of *V. vulnificus* [6] were utilized in this study (Table 1). The strains were routinely cultured on plates of TSA-1 (trypticase soy agar plus 5 g/L NaCl) or in tubes of LB-1 (Luria Bertani broth plus 5 g/L NaCl) at 28 °C for 24 h and were maintained in frozen stocks at −80 °C in LB-1-glycerol (ratio 1:5).

### 2.2. Electrolyzed Water (EW) Preparation and Characterizacion

EW was generated using a LAMI-50 generator (Aquactiva Solutions S.L., València, Spain), equipped with a membrane electrolysis cell. A solution of saturated NaCl and deionized water was utilized, eliminating the need for inlet water treatment. EW was characterized for pH, oxidation-reduction potential (ORP), and conductivity using the SG68 pH/Ion/DO Multimeter (Mettler Toledo, Columbus, OH, USA). Free available chlorine (FAC) was measured with a Handheld Colorimeter Chlorine UHR (Hanna Instruments, RI Woonsocket, Providence County, RI, USA). The physic-chemical parameters of EW tested in this work are shown in Table 2. Due to the instability of EW, all experiments were conducted immediately after its generation.

### 2.3. Bacterial Survival in EW

The bactericidal effect of EW on *V. vulnificus* was evaluated in microcosms under different conditions of salinity (3% and 1.5%), pH (5, 6.5, and 7.5), and FAC (5, 25, and 125 ppm) (Table 2) at 0-, 5-, 10-, and 15-min post-inoculation. The salinity and pH values were selected based on the average values in seawater, estuaries, and fish farms while the FAC values and action time were selected from the results obtained by other researchers [17]. To this end, overnight LB-1 cultures of *V. vulnificus* were diluted 1:100 in fresh LB-1 and then were inoculated (1:10 ratio) into freshly prepared EW (final volume 100 mL) and PBS (positive control) to achieve a bacterial population size of 10^6^ CFU/mL, the 50% lethal dose of this pathogen for bath-infected fish [6,18]. Survivors were then estimated by drop-counting on TSA-1 plates (limit detection 100 UFC/mL) [19]. The bactericidal effect of EW was only considered if survival percentage in the control after 15 min incubation was 100%. All experiments were conducted in triplicate.

### 2.4. Statistical Analysis

An analysis of variance of aligned rank transformed (ART ANOVA) was employed to test for significant differences in bacterial counts using R (version 4.3.2). When ART ANOVA indicated significance (*p* < 0.05), post hoc analyses were performed with *p*-values adjusted by Tukey’s method. Before proceeding with the analysis, normality was assessed using the Kolmogorov–Smirnov test. A normal distribution was found only in the data from the first analysis (*p* > 0.05). To evaluate the homogeneity of variances, Cochran’s test was applied, which showed no homoscedasticity in any of the data groups (*p* < 0.05). Based on these results, non-parametric tests were deemed appropriate. The effect sizes for different factors were calculated using partial eta squared (η^2^).

## 3. Results and Discussion

### 3.1. Influence of Salinity, pH and FAC on the Bactericidal Activity of EW

*V. vulnificus* is a bacterium adapted to survive in aquatic environments with a wide range of salinity (0.05–3.5%; optimal 1.5–2%) and pH (5–10; optimal 6–9), which theoretically allows it to survive and infect a broad range of aquaculture species [20]. In this study, we selected the conditions under which it infects in fish farms in our geographical area. Bacterial survival experiments with EW containing 5 ppm FAC (EW-5) showed that, at this chlorine concentration, EW was not bactericidal at any tested salinity and pH, with bacterial survival being 100% at all times tested In contrast, EW-125 was highly bactericidal regardless of salinity and pH, causing a population reduction of more than 4 log units in less than one minute. Both concentrations of FAC were discarded for further experiments, the former as ineffective and the latter because of its potential toxic effect on fish according to Kasai et al. [21]. This does not exclude the value of EW-125 as a powerful disinfectant for potential use in non-animal aquatic facilities, equipment, food products, or even fish farm effluent water [22].

Figure 1 shows the survival of strain CECT 4999 in EW-25 at 1.5 and 3% salt at the different pH values tested. Bacterial populations decreased significantly with incubation time in all conditions, although with differences depending on pH and salinity. Thus, at pH 5 and 7.5 the bactericidal effect at 5 and 15 min was greater than at pH 6.5, regardless of water salinity, apparently being faster and more intense in brackish water (1.5% salinity) at pH 5 (conditions used in some aquaculture facilities) and in salty water (3% salinity) at pH 7.5 (conditions close to seawater) (Figure 1). In fact, no colony was recovered from EW-25 at the two salinities and pH 5 as well as at 3% salinity and pH 7.5 at 15 min of incubation. These results show that EW-25 reduces bacterial population by more than 4 log units (Figure 1).

ART ANOVA analysis confirmed all the above observations, revealing that both pH and water salinity have a significant effect on bacterial survival, but salinity interactions do not (Table 3). Furthermore, time and its interaction with pH also have a significant effect. It should be noted that post hoc analysis of the data also revealed significant differences as a function of incubation time, except for the samples corresponding to times 0 and 1 min, suggesting that more than one minute of exposure is necessary for EW-25 to produce the bactericidal effect (Figure 1).

The partial eta squared value was then determined to ascertain the magnitude of the effect of each factor on the variance of the data. The time factor explained the greatest variance (80.79%), followed by pH (21.36%) and salinity (14.51%). Based on these results, we conclude that EW-25 can be bactericidal at any water pH and salinity as long as the time of action is adjusted to achieve the desired effect.

Figure 2 shows bacterial survival in EW adjusted to pH 5 at different salinity (0.5 and 1.5%) and FAC concentrations (15, 20, and 25 ppm) (EW-15, -20, and 25). As we expected, no differences were observed between salinities while the bactericidal effect was more rapid and intense at the highest FAC concentration (5 min, 2.5 log unit reduction), intermediate at the middle concentration (10 min, 2 log unit reduction), and low at the lowest concentration (15 min, 1 log unit reduction).

ART ANOVA statistical analysis revealed significant differences for the three factors and the three 2-to-2 interactions (Table 4), and post hoc analysis, significant differences as a function of time. Finally, partial eta squared analysis showed that exposure time explained the greatest variation (85.41%), followed by FAC (74.12%) and salinity (10.44%).

In summary, our results indicate that chlorine concentration and exposure time, rather than salinity or pH, are the key factors determining the bactericidal effectiveness of EW against *V. vulnificus*, as long as the values remain within the non-toxic range for fish, as tested in this study.

Of the three FAC concentrations tested, we discarded the lowest because of its poor efficacy and the highest because of its toxicity to animals [21]. Consequently, we selected EW-20 for the rest of the experiments. It has been suggested that similar concentrations may be non-toxic for different species of aquatic animals when used in aquaculture facilities [22].

### 3.2. Evaluation of the Bactericidal Effect on Different Strains of V. vulnificus

*V. vulnificus* is a genetically variable species, so we decided to evaluate the bactericidal power of EW-20 against strains representative of the five phylogenetic lineages described in the species by adjusting salinity to 0.5 and 1.5 (those corresponding to brackish water), and pH to 5 (Figure 3). For these experiments we selected parameters closely adjusted to the pH and salinity values used in fish farms in our geographical area.

The populations of the 5 strains decreased over time at the two salinities tested, but apparently with differences among them and between salinities (Figure 3). Thus, strains V12 (L3) and CECT 4999 (L2) were the most sensitive to both salinities while strains Riu1 (L4), V252 (L5), and YJ016 (L1) resisted the treatment better and strain VV5 gave different survival values according to salinity.

ART ANOVA analysis showed that water salinity did not significantly influence EW-20 efficacy, but sampling time did (Table 5). It also showed that the differences between strains were significant, confirming that intraspecific genetic differences affect the resistance of the isolate to the bactericidal action of EW-20. Although we do not yet know the genetic basis, the survival and resistance of *V. vulnificus* to various stressors depend on the production of a protective capsule, its serological type, and the post-transcriptional modification of LPS. These factors, which vary significantly between strains, could explain the observed variability [23].

Finally, the time parameter, again, explains the greatest variance in the data (83.13%), followed by the strain factor (48.65%) and salinity (2.35%). In any case, the reduction obtained in the population sizes of the five strains was significant, ranging from 1 to 2.5 log units at 15 min of incubation (Figure 3).

In summary, we have demonstrated the efficacy of EW against *V. vulnificus* at FAC concentrations of 20 ppm, and salinity and pH compatible with those used in fish farms where fish species susceptible to Vv-vibriosis are raised (around pH 5.5 and 0.5% salinity). There are other studies evaluating the efficacy of EW against different pathogens [17,24]. However, we cannot compare our results with those obtained in these studies because they used FAC concentrations and pH incompatible with animal life and initial bacterial concentrations far away from those found in aquaculture facilities (around 10^3−8^ CFU/mL). Finally, although the concentration of FAC that we selected is not toxic for multiple marine animals, we recommend confirming its non-toxicity for the target species before using EW treatment in aquaculture facilities.

## 4. Conclusions

This study demonstrates that EW is an effective disinfectant against the zoonotic pathogen *V*. *vulnificus*, regardless of lineage and strain: it significantly reduces *V*. *vulnificus* populations at FAC concentrations of 20 and 25 ppm and at water parameters compatible with the water of eel and tilapia farms, its main hosts (pH 5.5, salinity 0.5%). This treatment constitutes an environmentally friendly alternative to antibiotics therapy, as it would keep *V*. *vulnificus* populations under control in fish farms, reducing the probability of Vv-vibriosis outbreaks and, therefore, the use of antibiotics. Finally, from a "One Health" perspective, controlling Vv-vibriosis outbreaks in fish farms is essential not only for animal health, but also for human health, as it reduces the risk of transmission of *V. vulnificus* to humans, which is especially relevant as this pathogen is clearly expanding due to global warming.

## Figures and Tables

**Figure 1 microorganisms-12-01992-f001:**
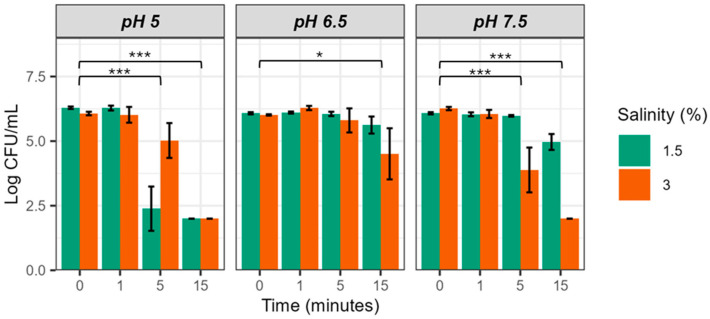
Survival of *V. vulnificus* in EW-25 (25 ppm of FAC) at different pH and salt concentrations. Strain CECT 4999 was used in this experiment. Values of bacterial counts below 10^2^ CFU/mL cannot be detected by drop-counting because of the detection limit of this method [19]. Significance codes: 0 (***) and 0.01 (*).

**Figure 2 microorganisms-12-01992-f002:**
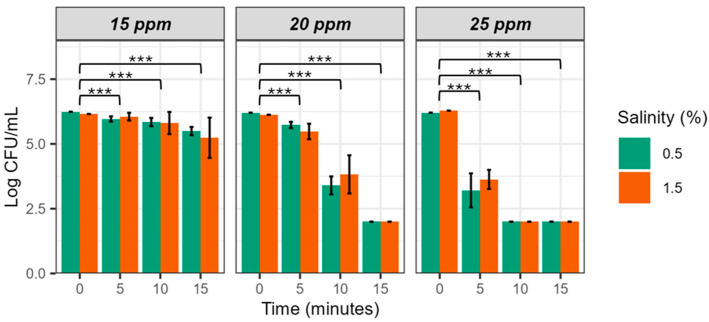
Survival of *V. vulnificus* in EW-pH5 and different FAC and salt concentrations. Strain CECT 4999 was inoculated in pH 5 EW at 15 ppm, 20 ppm, and 25 ppm of FAC. Values of bacterial counts below 10^2^ CFU/mL cannot be detected by drop-counting because of the detection limit of this method [19]. Significance code: 0 (***).

**Figure 3 microorganisms-12-01992-f003:**
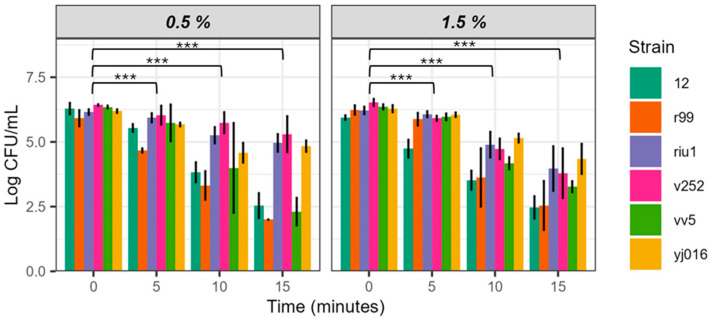
Survival of *V. vulnificus* in pH 5 EW-20 at different salt concentrations. Strains 12, CECT 4999, Riu1, V252, VV5, and YJ016 were inoculated pH 5 EW at 20 ppm of FAC. Values of bacterial counts below 10^2^ CFU/mL cannot be detected by drop-counting since this is the detection limit for this method [19]. Significance code: 0 (***).

**Table 1 microorganisms-12-01992-t001:** Characteristics of the *V. vulnificus* strains used in this study.

Strain ^1^	Source	Geographic Location	Year of Isolation	Lineage ^2^
CECT 4999	Diseased farmed eel	Spain	1999	L2
YJ016	Human blood	Taiwan	1993	L1
Riu1	Seawater	Western Mediterranean Sea	2003	L4
V252	Human blood	Israel	2004	L5
VV12	Human blood	Israel	1996	L3
VV5	Farmed diseased tilapia	Eastern Mediterranean Sea	2016	L1

^1.^ CECT: Spanish Type Culture Collection. ^2.^ L: Phylogenetic lineage [6].

**Table 2 microorganisms-12-01992-t002:** Physic-chemical parameters of EW tested in this study.

pH	FAC (ppm)	ORP (mV)	%NaCl	Conductivity (mS/cm)
7.5	5	828	1.5	26.2
		900	3	47.5
	25	889	1.5	25.7
		937	3	48.1
	125	943	1.5	25.9
		960	3	49.3
6.5	5	832	1.5	26.0
		902	3	47.4
	25	892	1.5	25.9
		933	3	47.9
	125	947	1.5	26.1
		962	3	49.5
5	5	828	1.5	26.3
		902	3	47.3
	15	835	1.5	26.1
		901	3	48.2
	20	890	1.5	26.2
		940	3	49.3
	25	890	1.5	26
		940	3	49.3
	125	950	1.5	26.2
		964	3	48.9

**Table 3 microorganisms-12-01992-t003:** Results of ART ANOVA analysis with 3 factors (salinity, pH and time) for CECT 4999 *V. vulnificus* survival in EW-25.

Source of Variation	Df ^1^	F ^1^	*p*-Value ^1^
Salinity	1	8.15	6.35 × 10^−4^ (**)
pH	2	6.52	3.13 × 10^−4^ (**)
Time	3	67.29	<2.2 × 10^−16^ (***)
Salinity X pH	2	2.56	0.087
Salinity X Time	3	2.07	0.117
pH X Time	6	8.69	2.04 × 10^−6^ (***)

^1.^ Df: degrees of freedom. F: F ratio. *p*-value: significance codes: 0 (***) and 0.001 (**).

**Table 4 microorganisms-12-01992-t004:** Results of ART ANOVA analysis with 3 factors (salinity, pH and time) for *V. vulnificus* survival (CECT 4999) in EW-pH5 at different FAC and salt concentrations.

Source of Variation	Df ^1^	F ^1^	*p*-Value ^1^
Salinity	1	5.596	2.2 × 10^−3^ (*)
FAC	2	68.75	8.12 × 10^−15^ (***)
Time	3	93.69	<2.2 × 10^−16^ (***)
Salinity X FAC	2	11.53	8.15 × 10^−5^ (***)
Salinity X Time	3	7.52	3.17 × 10^−5^ (***)
FAC X Time	6	20.17	1.31 × 10^−11^ (***)

^1.^ Df: degrees of freedom; F: F ratio. Significance codes: 0 (***) and 0.01 (*).

**Table 5 microorganisms-12-01992-t005:** Results of ART ANOVA analysis with 3 factors (salinity, pH and time) for different *V. vulnificus* strains in pH 5 EW-20 at different salt concentrations.

Source of Variation	Df ^1^	F ^1^	*p*-Value ^1^
Salinity	1	2.31	0.13
Strain	5	18.19	1.16 × 10^−12^ (***)
Time	3	157.69	<2.2 × 10^−16^ (***)
Strain X Salinity	5	8.74	7.33 × 10^−7^ (***)
Salinity X Time	3	4.89	3.3 × 10^−4^ (**)
Strain X Time	15	6.46	2.76 × 10^−9^ (***)

^1^ Df: degrees of freedom; F: F ratio. Significance codes: 0 (***) and 0.001 (**).

## Data Availability

The raw data can be found in Appendix A.

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
