# Peer review of "Electrolyzed Water Treatment for the Control of the Zoonotic Pathogen Vibrio vulnificus in Aquaculture: A One Health Perspective"

_microorganisms, 2024, doi:10.3390/microorganisms12101992_

Round 1
Reviewer 1 Report
Comments and Suggestions for Authors
Lines 10-12, 25-39: Explain how does Vibrio vulnificus act as a zoonotic pathogen, and what are its links for both human and animal health?
Lines 13-14, 60-67: What are the benefits of use electrolyzed water (EW) instead of conventional chemical disinfectants to manage Vibrio vulnificus in aquaculture settings?
Lines 110-125, 151-167: What is the relationship between the bactericidal activity of electrolyzed water against Vibrio vulnificus and pH, salinity, and free available chlorine (FAC) concentration?
Lines 143: What are the important variables that affect Vibrio vulnificus's ability to survive in various aquaculture environments?
Lines 174-191: Explain in brief about the influence of Vibrio vulnificus strain genetic variability on the efficiency of electrolyzed water treatment?
Author Response
Thanks for your review and your kind words. We have taken into account all your comments and add the following text in our revised MS (still working on it). We cut and paste the added text.
-------------------------------
Comment Lines 10-12, 25-39: Explain how does Vibrio vulnificus act as a zoonotic pathogen, and what are its links for both human and animal health?.
To clarify this, we have added at the end of the second paragraph in the introduction section page 1 the following text:
"Finally, there is clear evidence of transmission of the vibriosis from fish to humans after handling diseased fish in farms, so that controlling fish health has a direct impact on reducing the risk of transmission of the vibriosis to humans (Carmona-Salido et al., 2022)"
-----------------------------------------
Lines 13-14, 60-67: What are the benefits of use electrolyzed water (EW) instead of conventional chemical disinfectants to manage Vibrio vulnificus in aquaculture?
To clarify this, we have added the following text at the end of the Introduction section:
"Consequently, EW is a safe, eco-friendly, and cost-effective disinfectant with broad-spectrum antimicrobial action, low toxicity, and no harmful residues, making it superior to traditional methods and, therefore, adequate for aquaculture”.
---------------------------------------------
Lines 143: What are the important variables that affect Vibrio vulnificus's ability to survive in various aquaculture environments?
To clarify this, we have added the following text in point 3.1. first paragraph:
"V. vulnificus is a bacterium adapted to survive in aquatic environments with a wide range of salinity (0.05-3.5%; optimal 1.5-2) and pH (5-10; optimal 6-9), which theoretically allows it to survive and infect a broad range of aquaculture species [Deeb, R., Tufford, D., Scott, G.I. et al. Impact of Climate Change on Vibrio vulnificus Abundance and Exposure Risk. Estuaries and Coasts 41, 2289–2303 (2018). https://doi.org/10.1007/s12237-018-0424-5]. In this study, we only tested the conditions under which it infects in fish farms in our geographical area"
------------------------------------------------
Lines 110-125, 151-167: What is the relationship between the bactericidal activity of electrolyzed water against Vibrio vulnificus and pH, salinity, and free available chlorine (FAC) concentration?
To clarify this, we have added this text in page 6 second pargraph:
"Summarizing, our results indicate that chlorine concentration and exposure time, rather than salinity or pH, are the key factors determining the bactericidal effectiveness of EW against V. vulnificus, as long as these values remain within the non-toxic range for fish, as tested in this study"
Lines 174-191: Explain in brief about the influence of Vibrio vulnificus strain genetic variability on the efficiency of electrolyzed water treatment?
To clarify this, we have added this text in point 3.2., third paragraph.
"Although we do not yet know the genetic basis, the survival and resistance of V. vulnificus to various stressors depend on the production of a protective capsule, its serological type, and the post-transcriptional modification of LPS. These factors, which vary significantly between strains, could explain the observed variability (Pettis, G.S.; Mukerji, A.S. Structure, Function, and Regulation of the Essential Virulence Factor Capsular Polysaccharide of Vibrio vulnificus. Int. J. Mol. Sci. 2020, 21, 3259. https://doi.org/10.3390/ijms21093259)
We hope we have adequately addressed your questions and that by including the texts in the MS the reader will understand better our work
Reviewer 2 Report
Comments and Suggestions for Authors
The article entitled "Electrolyzed water treatment for the control of the zoonotic pathogen Vibrio vulnificus in Aquaculture: A One Health perspective" which was submitted to me for review is, in my opinion, extremely interesting and necessary. Currently, new environmentally friendly methods of combating infectious pathogens and preventing infectious diseases, including zoonoses, are being sought. Electrolyzed water (EW) is already successfully used in some branches of animal husbandry, for example in poultry farming. Attempts are also being made to use it in aquaculture (I even recently reviewed an article on this subject).
I have only one substantive comment because the article is very well written. I consider even the fact that it is relatively short to be its great advantage. However, I ask that the authors, when providing the NaCl concentration used, classify by specifying whether the water they used is fresh, brackish or salty. It is necessary to define when we are dealing with fresh, brackish and salty water, with reference to a good and reliable scientific source containing definitions.
The authors state "Homoscedasticity and normality were tested prior to analysis". They should add what statistical tests were used and what their results were. The mere statement that this was tested means absolutely nothing because we still do not know whether the assumptions for AVOVA were met or not. After all, they could have always been met, only sometimes, or they could not have been met at all. This should be clarified.
Author Response
Thanks for your comments. We have taken into account all of them.
Comment 1: I have only one substantive comment because the article is very well written. I consider even the fact that it is relatively short to be its great advantage. However, I ask that the authors, when providing the NaCl concentration used, classify by specifying whether the water they used is fresh, brackish or salty. It is necessary to define when we are dealing with fresh, brackish and salty water, with reference to a good and reliable scientific source containing definitions.
Response 1. Thank you for the suggestion. It certainly makes the work easier to follow by replacing the current nomenclature for the type of water used with 'fresh, brackish, or salty.' Therefore, we have followed your advice and made this change throughout the manuscript, where appropriate
Comment 2. The authors state "Homoscedasticity and normality were tested prior to analysis". They should add what statistical tests were used and what their results were. The mere statement that this was tested means absolutely nothing because we still do not know whether the assumptions for AVOVA were met or not. After all, they could have always been met, only sometimes, or they could not have been met at all. This should be clarified.
We have clarified your question in the Materials and Methods section. This is the text "Before proceeding with the analysis, normality was assessed using the Kolmogorov-Smirnov test. A normal distribution was found only in the data from the first analysis (p > 0.05). To evaluate the homogeneity of variances, Cochran's test was applied, which showed no homoscedasticity in any of the data groups (p < 0.05). Based on these results, non-parametric tests were deemed appropriate. The effect sizes for different factors were calculated using partial eta squared (η2)"